Relationship between ferroptosis and mitophagy in acute lung injury: a mini-review

Cheng Yunhua 1 2
Zhu Liling 3
Xie Shuangxiong 1 2
Lu Binyuan 1 2
Du Xiaoyu 4
Ding Guanjiang 1 2
Wang Yan 1 2
Ma Linchong 2
Li Qingxin 2 liqxchest@163.com
1 The First School of Clinical Medicine of Gansu University of Chinese Medicine, Gansu University of Chinese Medicine , Lanzhou, Gansu Province , China
2 Department of Thoracic Surgery, The 940th Hospital of Joint Logistics Support Force of Chinese People’s Liberation Army , Lanzhou, Gansu Province , China
3 Department of Anesthesiology, Hunan Children’s Hospital , Changsha, Hunan Province , China
4 Medical College of Northwest Minzu University, Northwest Minzu University , Lanzhou, Gansu Province , China
Eamens Andrew
Electronic publication date: 2024 Sep 10
Publication date: 2024
Volume: 12
Electronic Location ID: e18062
Received 2024 Mar 12; Accepted 2024 Aug 19
Copyright: © 2024 Cheng et al.
Copyright year: 2024
Copyright holder: Cheng et al.
License: This is an open access article distributed under the terms of the Creative Commons Attribution License, which permits unrestricted use, distribution, reproduction and adaptation in any medium and for any purpose provided that it is properly attributed. For attribution, the original author(s), title, publication source (PeerJ) and either DOI or URL of the article must be cited.
License URL: https://creativecommons.org/licenses/by/4.0/

Keywords: Acute lung injury, Ferroptosis, Mitochondria, Mitophagy, Relationship, Mechanism

Funding: Research Project of the 940th Hospital of Joint Logistics Support Force of Chinese People’s Liberation Army 2023YXKY030 This study was funded by the Research Project of the 940th Hospital of Joint Logistics Support Force of Chinese People’s Liberation Army (2023YXKY030). The funders had no role in study design, data collection and analysis, decision to publish, or preparation of the manuscript.

==============================
Acute lung injury (ALI) is one of the most deadly and prevalent diseases in the intensive care unit. Ferroptosis and mitophagy are pathological mechanisms of ALI. Ferroptosis aggravates ALI, whereas mitophagy regulates ALI. Ferroptosis and mitophagy are both closely related to reactive oxygen species (ROS). Mitophagy can regulate ferroptosis, but the specific relationship between ferroptosis and mitophagy is still unclear. This study summarizes previous research findings on ferroptosis and mitophagy, revealing their involvement in ALI. Examining the functions of mTOR and NLPR3 helps clarify the connection between ferroptosis and mitophagy in ALI, with the goal of establishing a theoretical foundation for potential therapeutic approaches in the future management of ALI.

Introduction

Acute lung injury (ALI), a prevalent and detrimental disorder of the respiratory system, is typified by neutrophil migration through the epithelial cell interstitium accompanied by an unchecked inflammatory response. This leads to substantial damage to lung epithelial cells and disruption of the cellular barrier, with the more severe stages evolving into acute respiratory distress syndrome (ARDS) (Butt, Kurdowska & Allen, 2016; Long, Mallampalli & Horowitz, 2022). Although earlier etiological control (Mokrá, 2020) and modified fluid management (Lee, Corl & Levy, 2021) have reduced its incidence, ALI continues to be associated with substantial morbidity and a high mortality rate. Accounting for over 10% of intensive care unit admissions and 4% of all hospitalizations, the elucidation of ALI and ARDS pathogenesis is pivotal to enhancing treatment for ALI (Meyer, Gattinoni & Calfee, 2021).

The primary function of the lungs is to carry out gas exchange, maintaining the ventilation-perfusion ratio within the normal range in the circulatory system. The energy required for this process is primarily supplied by the mitochondria (Hu & Königshoff, 2022). However, when mitochondrial function is impaired, it leads to an incomplete reduction of O2, subsequently producing reactive oxygen species (ROS) (Willems et al., 2015). Excessive ROS leads to a switch in cellular energy production from aerobic glycolysis to anaerobic glycolysis and pentose phosphate pathway, affecting the antioxidant capacity of the mitochondria (Schumacker et al., 2014). This further induces mitochondrial damage and the release of mitochondrial DNA (mtDNA), thereby activating inflammatory responses leading to ALI (Schumacker et al., 2014; Long et al., 2022). Persistent mitochondrial damage and dysfunction result in organ failure and poor outcomes in patients with ALI (Ten & Ratner, 2020). Therefore, the removal of unhealthy mitochondria and the generation of healthy mitochondria via mitochondrial quality control (MQC) are critically important for preserving the structural and functional integrity of mitochondria (Ng, Wai & Simonsen, 2021). Mitophagy, a selective autophagic process, is crucial for maintaining mitochondrial quantity, quality, and essential functions within cells (Ashrafi & Schwarz, 2013). This process facilitates the removal of damaged mitochondria and mitigates ROS production, thereby playing a vital role in the occurrence and development of ALI.

Ferroptosis is a newly recognized type of cell death that is involved in the onset of ALI (Liu, Zhang & Xie, 2022; Ma et al., 2023; Wang et al., 2023c). As an increasingly recognized therapeutic target, the inhibition of ferroptosis presents a promising strategy for mitigating ALI (Zhang et al., 2022). Despite growing interest, a thorough understanding of ferroptosis is still evolving. Ferroptosis is often accompanied by the accumulation of large amounts of iron ions and excessive lipid peroxidation products, characterized morphologically by alterations in the ultrastructure of mitochondria including reduced volume, heightened double-membrane density, and disruption of the outer mitochondrial membrane (Li et al., 2020b). During ferroptosis, the redox balance in the cell is disrupted, with levels of glutathione (GSH) and glutathione peroxidase 4 (GPX4) decreasing while ROS levels begin to increase (Jiang, Stockwell & Conrad, 2021). Mitochondria, as the main source of ROS in cells, are tightly associated to ferroptosis (Wang et al., 2020b; Javadov, 2022). Mitophagy facilitates the removal of damaged mitochondria, prevents mitochondrial dysfunction, and thus maintains normal mitochondrial morphology and functionality (Pickles, Vigié & Youle, 2018). However, the relationship between mitophagy and ferroptosis is currently unclear. Current research mainly focuses on the individual roles of ferroptosis and mitophagy in ALI (Liu, Zhang & Xie, 2022; Tang et al., 2023). However, there are fewer studies on the relationship between ferroptosis and mitophagy, especially their correlation in ALI is yet to be investigated. Consequently, in the context of ALI, exploring the direct or indirect interplay between mitophagy and ferroptosis is hypothesized to be fruitful, and dissecting their mechanisms could yield significant insights. In this review, we hypothesize that elucidating the relationship between mitophagy and ferroptosis will aid in finding effective treatment strategies for ALI. We aim to investigate and examine the disturbance mechanism between ferroptosis and mitophagy in ALI by exploring their interrelationship. This review delves into the novel interrelationship between mitophagy and ferroptosis in ALI, aiming to illuminate potential therapeutic avenues for this condition.

Survey methodology

The literature search was conducted in PubMed, Web of Science, and the China National Knowledge Infrastructure database. In addition to considering articles published since 2010, earlier articles were also taken into account. The following keywords were used: ALI, ARDS, ferroptosis, mitophagy, ALI and ferroptosis, ALI and mitophagy, iron metabolism, ferroptosis mechanisms, mitophagy mechanisms. As our work progressed, we conducted literature searches using the keywords NLRP3 and mitophagy, NLRP3 and ferroptosis, mTOR and mitophagy, mTOR and ferroptosis. After removing duplicate articles and the articles with little relevance, 126 articles were selected for inclusion in this review.

The rationale for why this review is needed

Numerous studies have extensively reviewed the roles of mitophagy and ferroptosis in various diseases, including acute kidney injury and myocardial ischemia-reperfusion injury. Furthermore, some researchers have comprehensively summarized the role of ferroptosis in ALI. However, to date, no scholar has summarized the link between mitophagy and ferroptosis in ALI. This review aims to provide the latest knowledge insights into the roles of mitophagy and ferroptosis in ALI, while exploring the more recently identified relationship between mitophagy and ferroptosis involved in ALI.

The audience this review is intended for

Doctors specializing in thoracic surgery, respiratory medicine and clinical laboratory medicine may find the review of current scientific literature of particular interest. The exploration of the interaction between mitophagy and ferroptosis contributes to a better understanding of the pathological process of ALI. Ultimately, with the continuous elucidation of the pathological changes of ALI and the application of recently developed therapeutic drugs, patients can expect better treatment and higher survival rates.

Acute lung injury and mitophagy

ALI is often accompanied by mitochondrial damage and dysfunction. To maintain homeostasis, lung tissue cells regulate the number of mitochondria they have to meet their physiological needs by undergoing processes like mitochondrial fission, fusion, and mitophagy (Dutra Silva et al., 2021). Normally, mitophagy is required to maintain cellular equilibrium through the decomposition and degradation of impaired mitochondria. Therefore, an elevated degree of mitophagy may result in mitochondrial malfunction, cellular impairment, and eventual cell death (Chan, 2020).

Regulation mechanism of mitophagy

Mitophagy regulation falls into two primary categories: non-receptor-mediated and receptor-mediated. Non-receptor-mediated mitophagy mainly involves the PINK1 (PTEN-induced kinase 1)/Parkin pathway, while receptor-mediated mitophagy includes proteins such as FUN14 domain containing 1 (FUNDC1), BCL2/adenovirus E1B 19 kDa interacting protein 3 (BNIP3), Bcl-2 homology 3 (BH3)-only protein Nix (BNIP3L), and Bcl2 family proteins. Should the PINK1/Parkin pathway be compromised, the receptor-mediated pathway may serve a compensatory function.

PINK1/Parkin and mitophagy

The PINK1/Parkin pathway is the foremost mechanism responsible for mediating mitophagy and represents a well-studied pathway in this context (Tanaka, 2020). PINK1, a serine/threonine kinase present on mitochondria, operates in concert with Parkin, an E3 ubiquitin ligase residing in the cytosol. In normal circumstances, PINK1 moves to the mitochondrial inner membrane (MIM) driven by the mitochondrial transmembrane potential (Δφ), and its mitochondrial targeting signal at the N-terminus is cut by the mitochondrial processing protease (MPP) (Fig. 1A) (Sekine & Youle, 2018). In the hydrophobic region of the inner membrane, the presenilin-associated rhomboid-like protein (PARL) cleaves PINK1 between amino acids Ala-103 (A103) and Phe-104 (F104), leading to the return of the residual PINK1 with F104 to the cytoplasm for rapid degradation via the ubiquitin proteasome system (UPS) (Deas et al., 2011). When mitochondria become damaged, the mitochondrial membrane potential decreases, preventing PINK1 from being degraded in the MIM. Consequently, PINK1 accumulates on the mitochondrial outer membrane (MOM), recruiting and phosphorylating Parkin to ubiquitinate various mitochondrial protein substrates (Kane et al., 2014). These ubiquitinated proteins bind to autophagy microtubule-associated protein 1, light chain 3 (LC3) to initiate mitophagy (Fig. 1B) (Geisler et al., 2010).

Figure 1 Mitophagy mechanism.

(A) Under physiological conditions, driven by the mitochondrial transmembrane potential, PINK1 is transported to the mitochondrial inner membrane and cleaved between amino acids A103 and F104 by PARL. Subsequently, PINK1 with the F104 residue returns to the cytoplasm and is degraded by the UPS. (B) When the mitochondria are damaged, Δφ decreases, preventing PINK1 from entering the inner membrane for degradation, and recruiting and phosphorylating Parkin, which ubiquitinates various mitochondrial protein substrates, thus initiating the autophagy process. (C) Src and CK2 enhance the interaction of FUNDC1 with LC3 to initiate mitophagy by dephosphorylating the Ser13 and Tyr18 sites of FUNDC1. The phosphorylation of Ser272 in BCL2L13 can promote its binding with LC3, thereby amplifying its role in inducing mitophagy. (D) Upon binding with the Atg8 protein family, BNIP3L localizes to the mitochondria through the TM domain, and the phosphorylation at the ser34/35 sites promotes interaction between LIR and LC3. The binding of BH3 with Bcl-2 leads to the disassociation of the Bcl-2/Beclin-1 complex, ultimately initiating mitophagy. JNK 1/2 phosphorylates the Ser 60/Thr 66 sites of BNIP3, promoting mitophagy, whereas PP1/2A dephosphorylates BNIP3, inhibiting mitophagy. (E) DRP1 mediates mitochondrial division into two mitochondria through its action on MFF, Fis-1, and Mid49/51, where the damaged mitochondrion is cleared by mitophagy. Abbreviations: TOM, the translocase of the outer membrane; Δφ, the mitochondrial membrane potential; MOM, mitochondrial outer membrane; MIM, mitochondrial inner membrane; PARL, the presenilin-associated rhomboid-like protein; MPP, mitochondrial processing peptidase; UPS, the ubiquitin proteasome system; Ub, ubiquitin; LC3, Autophagy microtubule-associated protein 1 light chain 3; Src, The sarcoma gene; CK2, Creatine kinase 2; FUNDC1, FUN14 domain containing 1; BCL2L13, Bcl2 like 13; LIR, LC3 interacting region; PP1/2A, Protein phosphatase 1/2 A; JNK1/2, C-Jun N-terminal kinase 1/2; BNIP3, Adenovirus E1B 19 kDa interacting protein 3; Atg8, Atg8-family proteins; TM, the transmembrane; BNIP3L, Bcl-2 homology 3 (BH3)—only protein Nix; BH3, the BH3 domain; MFF, Mitochondrial Fission Factor; Fis-1, Mitochondrial Fission 1 protein; Mid49/51, Mitochondrial Dynamics Proteins of 49 and 51 kDa. Created with BioRender.

FUNDC1 receptor and mitophagy

FUNDC1 is a MOM protein that harbors an LC3 interacting region (LIR) motif, enabling its interaction with LC3 amid hypoxic or mitochondrial stress conditions to facilitate mitophagy (Yang et al., 2019). Under normal physiological circumstances, FUNDC1 is present in a phosphorylated state. During hypoxic episodes, FUNDC1 is dephosphorylated, enhancing its binding to LC3 to promote mitophagy (Liu et al., 2012; Chen et al., 2014). This phosphorylation and dephosphorylation are regulated by the sarcoma gene (Src) kinase and creatine kinase 2 (CK2). Src kinase and CK2 can phosphorylate FUNDC1 at Ser13 and Tyr18 sites under normal physiological conditions, resulting in the formation of inactive p-FUNDC1 and the inhibition of mitophagy (Zheng et al., 2022). Under hypoxic conditions, Src kinase and CK2 remove phosphate groups from FUNDC1, increasing its binding with LC3 to start the process of mitophagy (Wang et al., 2020c). The stimulation of mitophagy via FUNDC1 is additionally mediated by ubiquitination. Under hypoxic conditions, the ubiquitin ligase MARCH5 catalyzes the ubiquitination of FUNDC1 at the Lys119 location, consequently driving the degradation of FUNDC1 (Fig. 1C) (Chen et al., 2017).

BCL-2 protein family and mitophagy

Members of the BCL-2 protein family are key regulators of cell apoptosis, with roles in regulating mitochondrial metabolism and dynamics. Bcl2 like 13 (BCL2L13) receptor, BNIP3, and BNIP3L/NIX, which are part of the BCL-2 protein family, have LIR domains that engage with LC3 to trigger mitophagy (Rogov et al., 2017; Li et al., 2020c). The mitophagy-inducing capacity of these proteins is modulated by their phosphorylation status. Phosphorylation of Ser272 in BCL2L13 enhances its binding to LC3, thereby strengthening its ability to induce mitophagy (Fig. 1C) (Murakawa et al., 2015). Under hypoxic conditions, c-Jun N-terminal kinase 1/2 (JNK 1/2) phosphorylates BNIP3 at Ser60/Thr66. This modification not only prevents the degradation of BNIP3 degradation by the proteosome, but also increases its stability, promotes LC3 interaction, and facilitates mitophagy. Conversely, dephosphorylation of BNIP3 by protein phosphatase 1/2 A (PP1/2A) leads to its proteasomal degradation, thereby impeding mitophagy (Fig. 1D) (He et al., 2022a). BNIP3L collaborates with the Atg8 family of proteins to direct autophagosomes to the intended mitochondria. BNIP3L/NIX primarily utilizes three domains to mediate mitophagy: LIR, the transmembrane (TM) domain and the BH3 domain. The transmembrane domain ensures localization of BNIP3L to the mitochondria, while the interplay between the LIR domain and LC3 can instigate mitophagy. Phosphorylation of NIX at Ser34/35 can augment mitophagy by promoting the interaction between the LIR and LC3, in addition to attracting mitochondria-specific autophagosomes (Rogov et al., 2017). Moreover, the BH3 domain at the NIX amino terminal holds the capacity to associate with Bcl-2, causing the dissociation of the Bcl-2/Beclin-1 complex and subsequent release of Beclin-1, thereby facilitating autophagosome formation and initiating mitophagy (Fig. 1D) (Bellot et al., 2009).

Mitochondrial fission and fusion and mitophagy

Mitochondria are dynamic organelles undergoing constant fission and fusion. Mitochondrial fission plays a crucial role in quality control, as it can divide damaged mitochondria into two: one with normal function and one with impaired function (Ni, Williams & Ding, 2015). DRP1 mediates mitochondrial fission through its action on mitochondrial fission factor (MFF), Mitochondrial Fission 1 protein (Fis-1), and mitochondrial dynamics proteins of 49 and 51 kDa (Mid49/51) (Losón et al., 2013). Subsequently, mitophagy maintains mitochondrial homeostasis by removing dysfunctional mitochondria (Fig. 1E) (Meyer, Leuthner & Luz, 2017). An imbalance in the processes of fission and fusion—a condition that results in mitochondrial fragmentation—serves as a prerequisite for mitophagy (Chan, 2020). Mitophagy is facilitated by mitochondrial fission, and conversely, suppression of mitophagy is achieved through the inhibition of mitochondrial fission or the stimulation of mitochondrial fusion (Wang et al., 2020a).

The role of mitophagy in acute lung injury

Numerous research studies have shown a connection between mitophagy and the onset and progression of ALI. The involvement of mitophagy in ALI is still a topic of debate, as studies have shown conflicting evidence regarding the impact of mitophagy on ALI (Ornatowski et al., 2020; Mohsin et al., 2021). Critical for cellular balance, mitophagy degrade and clear damaged mitochondria; however, if excessive, it can precipitate mitochondrial dysfunction, cellular impairment, and apoptosis. In cecal ligation and puncture (CLP) and lipopolysaccharide (LPS)-induced ALI, mitophagy results in the elimination of damaged mitochondria, reducing ROS production and thereby alleviating ALI (Sang et al., 2022). Conversely, studies have suggested that alleviating ALI can be achieved by inhibiting mitophagy to reduce ROS and inflammatory substance production (Xiao et al., 2023). Numerous studies have probed into understanding the role of mitochondrial malfunction in the pathogenic process underlying ALI. Mitophagy plays different roles in ALI in different cell types. Studies have shown that in mouse models of ALI, mitophagy is primarily present in macrophages and alveolar epithelial cells (Chang et al., 2015).

Research has shown that mitophagy within alveolar macrophages can mitigate the severity of ALI. Specifically, mitophagy appears to modulate the activation of the NLRP3 inflammasome in these cells (Zhou et al., 2011). The highly conserved Sestrin2 (Sesn2) protein exerts a protective effect in LPS-induced ALI by promoting mitophagy to protect alveolar macrophages and reduce the release of NLRP3 inflammasomes (Wu et al., 2021). In contrast, the antioxidant mitochondria-targeted antioxidant mitoquinone (MitoQ) improves ALI by inhibiting mitophagy in macrophages and decreasing NLRP3 inflammasome activation (Sang et al., 2022). White et al. (2022) have observed that cigarette smoke intensifies Pseudomonas aeruginosa-induced mitophagy, resulting in an accumulation of p62, exacerbation of mitochondrial damage, and heightened activation of NLRP3 inflammasome in alveolar macrophages, which ultimately aggravates Pseudomonas aeruginosa-induced ALI. Consequently, it underscores the necessity for further investigation into the precise regulatory mechanisms of mitophagy on NLRP3 inflammasomes in macrophages.

Mitophagy in alveolar epithelial cells also has varying effects on ALI. Damaged mitochondria release ROS and apoptotic factors, leading to cell death or apoptosis by activating the mitophagy pathway in alveolar epithelial cells (Tian et al., 2022). Inhibition of mitophagy within alveolar epithelial cells may mitigate ALI. In type II alveolar cells, histone deacetylase 3 (HDAC3) enhances the transcription of Rho-associated protein kinase 1 (ROCK1), which is phosphorylated by Rho-associated (RhoA) following LPS stimulation. HDAC3 also diminishes the acetylation level of FOXO1, a transcription factor of ROCK1, thereby promoting mitophagy via the FOXO1-ROCK1 axis and contributing to ALI (Li et al., 2023). Under high oxygen conditions, the expression of BMI1 (B cell-specific Moloney murine leukemia virus integration site 1) significantly decreases, resulting in loss of Δφ, increased expression of PTEN (phosphatase and tensin homolog) and Pink1, enhanced mitophagy, and cell death in alveolar epithelial cells, exacerbating ALI. Conversely, mitophagy can offer protection against ALI. Zhuang et al. (2021) discovered that MCTR3 obstructs the ALX/PINK1 pathway in lung cells, diminishing mitophagy and lessening the severity of LPS-induced ALI. Furthermore, overexpression of PGC-1α (PPARγ coactivator 1α) elevates transcription factor EB (TFEB) expression, facilitating mitophagy in alveolar epithelial cells to ameliorate LPS-induced ALI (Liu et al., 2019b, 2021a).

Melatonin, a key hormone produced by the pineal gland, has shown strong antioxidant and anti-inflammatory effects that can be helpful in managing heart and lung diseases (Ling et al., 2023b). This hormone modulates mitophagy and influences the regulation of inflammatory cytokines. By inhibiting Optineurin (OPTN)-related mitophagy via the PINK1/Parkin pathway and suppressing STAT3 (signal transducer and activators of transduction 3) and TNF-α (tumour necrosis factor alpha) expression through the JAK2/STAT3 pathway, melatonin effectively mitigates ALI. Therefore, the involvement of melatonin in controlling mitophagy via the JAK2/STAT3 pathway helps decrease excessive mitophagy, suggesting potential therapeutic benefits in reducing the severity of ALI (Ning et al., 2022; Ling et al., 2023b).

Acute lung injury and ferroptosis

Ferroptosis, first identified in RAS mutant tumor cells upon erastin exposure, exhibits unique morphological characteristics distinct from apoptosis, necrosis, and other forms of cell death (Dolma et al., 2003). This process is marked by mitochondrial shrinkage, diminution or loss of mitochondrial cristae, condensed mitochondrial membranes, and moderate chromatin condensation (Li et al., 2020b). Subsequent research has elucidated the role of ferroptosis in the pathogenesis of a myriad of diseases, including cancer (Lei, Zhuang & Gan, 2022), lung (Wang et al., 2022a) and kidney disorders (Wang et al., 2022b), as well as myocardial ischemia-reperfusion injury (Cai et al., 2023). In the context of pulmonary conditions, ferroptosis has been linked to ALI triggered by various factors such as infection, radiation, ischemia-reperfusion, drowning, and oleic acid exposure (Table 1). In ALI, disturbed iron homeostasis leads to iron accumulation in the lower respiratory tract, precipitating ferroptosis, which, in concert with oxidative stress and mitochondrial dysfunction, aggravates lung damage (Fig. 2).

Table 1 ALI and ferroptosis.

ALI	Manifestations	Pathway	
SRALI	The representation of SLC7A11 and GPX4 experienced a decrease concurrently with an escalation in the levels of MDA and overall iron content.	P53/SLC7A11 (Chen et al., 2022a, 2022b, 2023a; Cao et al., 2023)	
Sirt1/Nrf2/Gpx4 (Deng et al., 2023; Lin et al., 2023b)	
Nrf2, ATF3 (Wang et al., 2022a)	
Keap1-Nrf2-GPX4 (Wang et al., 2022b; Shen et al., 2023)	
Nrf2/HO-l (Tang et al., 2022; Wang et al., 2023b)	
Keap-Nrf2-HO-l (Li et al., 2021; Lou et al., 2023)	
Nrf2 (He et al., 2022b; Li et al., 2022b)	
Ca2+ (Cai et al., 2022)	
mTOR (Li et al., 2022b)	
Srg3 (Ling et al., 2023a)	
hippo (Wang et al., 2022c)	
MAPK/ERK (Wang et al., 2022d)	
STAT6-P53/SLC7A11 (Yang et al., 2022)	
RRALI	Lowered GPX4 expression triggers excess ROS, impairs the alveolar endothelial barrier.	p62-keap1-NRF2 (Li et al., 2022a)	
IRRALI	A downtrend in the expression of GPX4 and GSH content leads to an uptick in MDA and lipid peroxidation levels.	Nrf2/TERT/SLC7A11 (Dong et al., 2021)	
Nrf2/HIF-l/TF (Li et al., 2020a)	
HIF-1 (Zhongyin et al., 2022)	
Nrf2/HO-1 (Tang et al., 2022)	
Nrf2/STAT3 (Qiang et al., 2020)	
PPARγ (Lu et al., 2023)	
PI3K/akt/mTOR (Zhang et al., 2023)	
DRALI	The expression of GSH decreases, leading to an increase in the levels of ROS and lipid peroxidation.	Nrf2 (Qiu et al., 2020)	
TNFAIP3-ACSL4 (Ling, 2022)	
OARALI	The levels of GSH, GPX4, and ferritin were decreased, and MDA levels were increased.	NLRP1 (Chen et al., 2019)	
Note:

Abbreviations: ALI, acute lung injury; SRALI, sepsis related acute lung injury; RRALI, radiation related acute lung injury; IRRALI, Ischemia-reperfusion related ALI; DRALI, drowning related acute lung injury; OARALI, oleic acid related acute lung injury; SLC7A11, solute carrier family 7, membrane 11; GPX4, glutathione peroxidase 4; MDA, malondialdehyde; ROS, acute respiratory distress syndrome; GSH, glutathione.

Figure 2 Regulation of key factors in ferroptosis and the treatment of ferroptosis antagonists in the ALI model.

The key factors regulating ferroptosis in various types of ALI, as well as the therapeutic effects of using ferroptosis antagonists on ALI models. Stat6 inhibits acetylation of P53, thereby restoring transcription of SLC7A11 and GPX4. GPX4 further reduces the levels of ROS, alleviating cellular damage caused by ROS, in turn, inhibiting ferroptosis. Nrf2 has several ways to regulate ferroptosis in ALI: Nrf2 stimulates the expression of STAT3, thereby upregulating SLC7A11 and inhibiting ferroptosis; Nrf2 facilitates the expression of HO-1 and ARE, restraining ferroptosis. Furthermore, Nrf2 can also suppress the activity of HIF -1α, promoting the expression of GPX4, and hence inhibiting ferroptosis. Abbreviations: SRALI, sepsis related acute lung injury; RRALI, radiation related acute lung injury; IRRALI, Ischemia-reperfusion related ALI; DRALI, drowning related acute lung injury; OARALI, oleic acid related acute lung injury; P53, tumor protein P53; STAT6, signal transducer and activators of transduction 6; SLC7A11, solute carrier family 7, membrane 11; GPX4, glutathione peroxidase 4; ROS, acute respiratory distress syndrome; STAT3, signal transducer and activators of transduction 3; HIF-1α, hypoxia induicible factor-1 alpha; TF, tissue factor; HO-1, heme oxygenase-1; Nrf2, nuclear factor erythroid 2-related factor 2; ARE, antioxidant response element; TNFAIP3, tumor necrosis factor alpha-induced protein 3; ACSL4, acyl-CoA synthetase long-chain family member 4; ALI, acute lung injury. Created with BioRender.

Ferroptosis plays a crucial role in the occurrence and progression of sepsis related ALI (SRALI). In a mouse model of ALI caused by LPS, there is a notable rise in the levels of unbound iron in the bronchial epithelial cells of ALI mice, accompanied by a significant decrease in the levels of ferroptosis indicators GPX4 and solute carrier family 7, membrane 11 (SLC7A11). Administering ferrostatin-1 as a preventive measure significantly reduced the severity of ALI, highlighting the crucial involvement of ferroptosis in the development of LPS-induced ALI. Additionally, research has shown that the P53/SLC7A11 and Nrf2/ARE pathways could play a role in controlling ferroptosis in LPS-induced ALI. It has been observed that STAT6 can suppress ferroptosis and reduce ALI by adjusting the P53/SLC7A11 pathway (Yu et al., 2014; Liu et al., 2020; Yang et al., 2022). Ferroptosis also plays a significant role in radiation related ALI (RRALI). In an RRALI mouse model, a decrease in GPX4 expression—a marker of ferroptosis—and mitochondrial alterations suggestive of ferroptosis were visible under transmission electron microscopy. Following treatment with an ferroptosis inhibitor, levels of lung ROS and serum inflammatory factors (TNF-α, IL-6, IL-10, and TGF-β1) significantly decrease, indicating the critical role of ferroptosis in RRALI (Li, Zhuang & Qiao, 2019). Li et al. (2022a) have revealed that activating the P62-Keap1-Nrf2 pathway could prevent radiation-induced ferroptosis in RRALI.

In ischemia-reperfusion related ALI (IIRALI), the cellular tumor antigen p53 modulates disease progression by curbing apoptosis and ferroptosis. Li et al. (2020a) showed that the inhibitor of apoptosis stimulating p53 protein (iASPP) can suppress ferroptosis and alleviate ALI induced by intestinal ischemia-reperfusion in mice by mediating the Nrf2/HIF-1/TF pathway (Li et al., 2020a). Additionally, isoliquiritigenin shields against this ALI by diminishing HIF-1-associated ferroptosis (Zhongyin et al., 2022). In cases of drowning related ALI (DRALI), employing an Nrf2-specific activator, such as dimethyl fumarate, results in the reduction of ROS and lipid ROS levels, a rise in GPX4 mRNA levels, and preservation of Δφ. Additionally, Nrf2 knockout mice show more pronounced lung damage than wild-type mice, suggesting that Nrf2 can suppress ferroptosis and mitigate ALI caused by seawater exposure (Qiu et al., 2020). Ling (2022) further showed that in seawater drowning-induced ALI, SOX9 facilitates lung epithelial cell ferroptosis via the TNFAIP3-ACSL4 pathway. In a murine model of oleic acid-related ALI (OARALI), reduced levels of GPX4 and ferritin are observed along with GSH depletion and accumulation of malondialdehyde in lung tissues, indicating the presence of ferroptosis (Chen et al., 2019; Zhou et al., 2019). These discoveries underscore ferroptosis as a key factor in ALI pathogenesis and a potential therapeutic target, warranting deeper examination into its specific pathways.

Ferroptosis and mitophagy in acute lung injury

The pathogenesis of ALI is closely related to mitophagy and ferroptosis, indicating potential interactions between them. Recent studies have unveiled significant interactions between ferroptosis and mitophagy: mitophagy can regulate the process of cellular ferroptosis (Table 2) (Bi et al., 2024). When cells undergo pathological changes, dysfunctional mitochondria produce excessive ROS and release pro-apoptotic factors, leading to further cell damage. Therefore, removal of dysfunctional mitochondria is essential for cellular homeostasis and survival. Ferroptosis leads to pathological alterations in cells, and mitophagy can clear dysfunctional mitochondria (Liu et al., 2023). Interestingly, similar to the role of mitophagy in ALI, it can both augment and suppress ferroptosis. In the early stage, mitophagy may sequester iron in autophagosomes, reducing the source of ROS in ferroptosis. However, in the massive mitophagy stage, it could provide additional iron, thus increasing lipid peroxidation and ferroptosis (Yu et al., 2022). Therefore, depending on the degree of mitophagy, its effects on ferroptosis may differ.

Table 2 Mitophagy and ferroptosis.

Mitophagy mechanism	Manifestations	Pathway	
PINK1	Early stage: ROS reduction and ferroptosis inhibition.
Large-scale mitophagy stage: Promotion of ferritinophagy, increase of iron ions, enhancement of lipid peroxidation and ferroptosis.	PINK1-PARK2/ROS/HO-1/GPX4 (Lin et al., 2023a)	
	Melatonin/PINK1/ROS (Zhou et al., 2023)	
	UHRF1/TXNIP/PINK1/ferroptosis (Ji et al., 2024)	
	SIRT1-SIRT3/PINK1/ferroptosis (Liao et al., 2023)	
	PINK1/de-O-GlcNAcylation/mitophagy/ferritinophagy/Fe2+/ROS/ferroptosis (Yu et al., 2022)	
	PINK1/HO-1/ferroptosis (Li et al., 2024)	
	PINK1/PARKIN (Qian et al., 2022)	
FUNDC1	FUNDC1/ACSL4 (Pei et al., 2021)	
	FUNDC1/GPx4/TOM/TIM (Bi et al., 2024)	
BNIP3	BNIP3/ROS/HO-1/GPX4 (Lin et al., 2023a)	
	BNIP3/mtROS (Yamashita et al., 2024)	
Note:

Abbreviations: PINK, PTEN-induced kinase 1; FUNDC1, FUN14 domain containing 1; BNIP3, Adenovirus E1B 19 kDa interacting protein 3.

Ferroptosis and mitophagy can interact through multiple network nodes, which can in turn influence each other and play unique roles. For instance, O-GlcNAcylation (a major nutrient sensor of the glucose flux) can regulate mitophagy and ferroptosis through ferritinophagy. Yu et al. (2022) found that O-GlcNAcylation promotes ferritinophagy, regulates mitophagy, and thereby controls ferroptosis. When ferroptosis inducers induce cell ferroptosis, the O-glucosyltransferase is deactivated, leading to de-O-GlcNAcylation, consequently activating ferritinophagy and mitophagy; ferritinophagy and mitophagy together provide ferrous ions, leading to the rapid generation of ROS and lipid peroxidation, ultimately resulting in ferroptosis. Moreover, inhibiting mitophagy by knocking down PINK1 at least partially prevents de-O-GlcNAcylation-promoted ferroptosis, and simultaneously inhibiting these two pathways nearly completely prevents ferroptosis (Yu et al., 2022). Song et al. (2024) also found ferritinophagy and mitophagy to play a crucial role in cell ferroptosis. In the process of Hexavalent chromium-induced cell death, ferritinophagy increases, disrupting iron homeostasis and releasing ferrous ions. In addition, hexavalent chromium triggers mitophagy, thereby releasing additional ferrous ions. These two pathways induce the simultaneous release of ferrous ions, thereby intensifying lipid peroxidation and ultimately triggering ferroptosis in DF-1 cells (Song et al., 2024).

In conclusion, there are close connections between mitophagy and ferroptosis, and a deeper exploration of the related mechanisms can provide new insights for research on ALI, which is closely related to mitophagy and ferroptosis. However, the relationship between mitophagy-related ferroptosis and ALI has not yet been studied. We aim to identify a common network node among these three by analyzing the relationships between mitophagy and ALI, and ferroptosis and ALI, as well as the regulatory mechanism of mitophagy on ferroptosis. From a mechanistic perspective, we found that both mTOR and NLRP3 play significant roles in the regulation of mitophagy and ferroptosis (Fig. 3).

Figure 3 Mitophagy and ferroptosis in ALI.

Ferroptosis plays a significant role in the pathogenesis of ALI, with mitophagy serving a dual regulatory function in both. Mitophagy can inhibit ROS and NLRP3, alleviating ferroptosis and ALI, but also has the potential to release active iron ions through ferritinophagy, thereby exacerbating ferroptosis and ALI. MTOR can inhibit ferroptosis and exert a dual regulatory role in mitophagy and ALI. Abbreviations: NLRP3, NOD-like receptor protein 3; ROS, reactive oxygen species; ALI, Acute Lung Injury. Created with BioRender.

The role of NLRP3 in ferroptosis and mitophagy

The protein NLRP3 has a significant association with ferroptosis, predominantly due to alterations in ROS levels. Operating as a key regulator of innate immunity and inflammatory reactions, the NLRP3 inflammasome initiates the activation of caspase-1, culminating in the subsequent secretion of pro-inflammatory cytokines like IL-1β and IL-18 in response to danger signals or the presence of pathogens (Kelley et al., 2019). In a sepsis related ALI murine model, created via CLP, markers of oxidative stress and ferroptosis were notably altered, with heightened NLRP3 expression, malondialdehyde, ROS levels, 4-hydroxy-2-nonenal (4-HNE) proteins, and iron accumulation observed alongside a notable reduction in GPX4 expression (Cao et al., 2022). In studies using mice to simulate PM2.5-induced lung damage and in cell models of lung epithelial cells, there was worsened lung tissue damage, increased levels of inflammatory mediators and NLRP3 protein, and signs of increased lipid peroxidation and iron buildup, along with reduced GPX4 expression and heightened ferroptosis, all of which were further impacted by ROS (Wang et al., 2023a). Thus, it is clear that the effect of NLRP3 on ferroptosis is primarily mediated through ROS modulation, with GPX4 playing a pivotal role in attenuating ROS and thereby mitigating ferroptosis.

The NLRP3 inflammasome is implicated in a spectrum of lung diseases including ALI, chronic obstructive pulmonary disease, lung cancer, pulmonary fibrosis, and various lung infections (Chen et al., 2023b). Reports suggest that ROS can promote the production of NLRP3, and ROS production can also be promoted by NLRP3. Mitophagy allows for the elimination of damaged mitochondria and leads to the reduction of ROS production, in turn inhibiting NLRP3 inflammasome activation (Kim, Yoon & Ryu, 2016; Mangan et al., 2018; Wu & Cheng, 2022). Therefore, NLRP3 gene is a significant target for ALI treatment. NLRP3 is closely related to PINK1/Parkin-mediated mitophagy, which helps in preventing and treating ALI (Zhang et al., 2014). Normal mitochondria translocate PINK1 to the inner mitochondrial membrane for degradation. When mitochondrial function is compromised, such as through membrane depolarization, dysfunction of complexes, or stress related to mutations, there is an increase in the accumulation of PINK1 on the damaged outer membrane of the mitochondria. Following this, the process triggers the mobilization and stimulation of Parkin from the cytoplasm. As a result of Parkin activation, mitochondrial membrane proteins are ubiquitinated, allowing them to be recognized by autophagosomes and then lysed by lysosomes (Li et al., 2021). While this PINK1-mediated mitophagy is documented to suppress NLRP3 inflammasome activation in acute liver ischemia-reperfusion injury, its role in regulating NLRP3 in the context of ALI, though unreported, is presumed to be of significant therapeutic relevance (Shan et al., 2019).

The role of mTOR in ferroptosis and mitophagy

As a serine/threonine kinase, mTOR consists of mTORC1 and mTORC2, playing important regulatory roles in cell growth, metabolism, and ferroptosis. In tumor cells, suppressing mTOR can lead to GPX4 degradation and promote ferroptosis (Liu et al., 2021b; Ni et al., 2021). Notably, in an ALI paradigm instigated by CLP, a surge in mTOR expression coupled with augmented mitophagy has been observed (Li et al., 2022b). Conversely, in an ischemia-reperfusion mouse model, mTOR activation mitigates ALI by impeding ferroptosis (Zhang et al., 2023). The precise role of mTOR in ferroptosis remains elusive; however, emerging evidence underlines its undeniable influence on the process and its link to mitophagy (Kang et al., 2011). Autophagy is known to be regulated via an mTOR-dependent pathway, with further studies substantiating the profound effects of mTOR and mitophagy on lung disease pathogenesis, including ALI. Liu et al. (2019a) found that the mTOR pathway is involved in the regulation of mitophagy in I/R and H/R-mediated ALI, modulating cell apoptosis. Inhibition of mTOR can induce mitophagy, enhance cell apoptosis, and exacerbate lung injury. On the other hand, Zhong et al. (2023) found that mTOR is also associated with mitophagy in LPS-induced ALI, but their results are inconsistent with those of Liu et al. (2019a) as they showed that inhibition of mTOR also suppresses mitophagy (Zhong et al., 2023). Controversies exist regarding the research results on whether mTOR promotes or inhibits mitophagy to alleviate ALI. This involves the issue of excessive mitophagy. In ALI, due to various factors such as the type of ALI, ischemia, and reperfusion time, mitophagy has a dual role in inflammation, cell apoptosis, and other processes. In ischemia-reperfusion-mediated ALI, mitophagy can inhibit cell apoptosis at the beginning of lung ischemia, but when mitophagy reaches a normal level, uncontrolled cell apoptosis exacerbates lung injury. Under certain circumstances, an excess of mitophagy, often coinciding with intensified autophagy, initiates the degradation of ferritin, causes the accumulation of ROS, ultimately leading to ferroptosis. Therefore, it is beneficial to overexpress mTOR by inhibiting mitophagy and ferroptosis when mitophagy is excessive. In the early stages of ALI, hypoxic conditions and energy deficits may hinder mTOR function and activate mitophagy; however, inhibiting mTOR can serve to exacerbate ferroptosis. Based on our preceding hypothesis that identifies ferroptosis in ALI as being largely dependent on mitochondrial ROS, despite the repression of mTOR expression, the initiation of mitophagy can reduce ROS generation by inhibiting the Fenton reaction, thus attenuating ferroptosis. Unfortunately, clear parameters to assess the extent of mitophagy in current ALI animal and cellular models remain unclear. We posit that a comprehensive investigation is imperative to elucidate the involvement of mTOR in reducing ALI.

Conclusions

In this review, we report that both mitophagy and ferroptosis are closely related to the onset and development of ALI. Mitophagy degrades damaged or dysfunctional mitochondria through multiple regulatory mechanisms, and in turn maintains the quality control of mitochondria and regulates ROS released by damaged mitochondria, thus playing an important role in ALI. Secondly, it is worth noting that ferroptosis is also closely related to ROS, mitophagy and ALI, and studies on both provide potential therapeutic options for ALI. However, current research is still insufficient, as the functions of mitophagy vary under different conditions in relation to ALI and ferroptosis, possibly depending on its regulatory mechanisms, extent and different causes of injury, further in-depth research is needed. Lastly, current studies often focus on either inhibiting ferroptosis or independently regulating mitophagy; therefore, the mutual regulation mechanisms between mitophagy and ferroptosis in ALI need further exploration. Through our investigation and analysis, in ALI, NLRP3 and mTOR play a significant ‘bridging’ role between ferroptosis and mitophagy, providing promising new targets for in-depth studies on the molecular mechanisms and interactions involved in these two processes. In summary, there is a close relationship between mitophagy, ferroptosis and ALI, with NLRP3 and mTOR acting as key nodes within these relationships. A deeper understanding of these regulatory processes could provide new insights into ALI research.

Supplemental Information

Supplemental Information 1 Response letter (adding a new author).

The technical assistance of BioRender is greatly acknowledged.

Abbreviations

ALI Acute lung injury

ROS Reactive oxygen species

ARDS Acute respiratory distress syndrome

mtDNA Mitochondrial DNA

MQC Mitochondrial Quality Control

GSH Glutathione

GPX4 Glutathione peroxidase 4

PINK1 PTEN-induced kinase 1

FUNDC1 FUN14 domain containing 1

BNIP3 Adenovirus E1B 19 kDa interacting protein 3

BNIP3L Bcl-2 homology 3 (BH3)-only protein Nix

MIM Mitochondrial inner membrane

MPP Mitochondrial processing protease

PARL Presenilin-associated rhomboid-like protein

UPS Ubiquitin proteasome system

Δφ The mitochondrial membrane potential

MOM Mitochondrial outer membrane

LC3 Autophagy microtubule-associated protein 1 light chain 3

LIR LC3 interacting region

Src The sarcoma gene

CK2 Creatine kinase 2

BCL2L13 Bcl2 like 13

JNK 1/2 C-Jun N-terminal kinase 1/2

PP1/2A Protein phosphatase 1/2 A

TM The transmembrane domain

BH3 Bcl-2 homology 3

MFF Mitochondrial Fission Factor

Fis-1 Mitochondrial Fission 1 protein

Mid49/51 Mitochondrial Dynamics Proteins of 49 and 51 kDa

CLP Cecal ligation and puncture

LPS Lipopolysaccharide

Sesn2 Sestrin2

MitoQ Mitoquinone

HDAC3 Histone deacetylase 3

ROCK1 Rho-associated protein kinase 1

RhoA Rho-associated

BMI B cell-specific Moloney murine leukemia virus integration site 1

PTEN Phosphatase and tensin homolog

MCTR3 Maresin conjugates in tissue regeneration

PGC-1a PPARγ coactivator 1a

TFEB Transcription factor EB

OPTN Optineurin

STAT3 Signal transducer and activators of transduction 3

TNF-α Tumour necrosis factor alpha

SRALI Sepsis-related ALI

GPX4 Peroxidase 4

SLC7A11 Solute carrier family 7, membrane 11

RRALI Radiation-related ALI

IRRALI Ischemia-reperfusion-related ALI

IASPP Inhibitor of apoptosis stimulating p53 protein

DRALI Drowning-related ALI

OARALI Oleic acid-related ALI

4-HNE 4-hydroxy-2-nonenal

Additional Information and Declarations

Competing Interests

Author Contributions

Data Availability

The authors declare that they have no competing interests.

Yunhua Cheng conceived and designed the experiments, performed the experiments, analyzed the data, prepared figures and/or tables, authored or reviewed drafts of the article, and approved the final draft.

Liling Zhu performed the experiments, analyzed the data, prepared figures and/or tables, authored or reviewed drafts of the article, and approved the final draft.

Shuangxiong Xie analyzed the data, authored or reviewed drafts of the article, and approved the final draft.

Binyuan Lu analyzed the data, authored or reviewed drafts of the article, and approved the final draft.

Xiaoyu Du analyzed the data, authored or reviewed drafts of the article, and approved the final draft.

Guanjiang Ding analyzed the data, authored or reviewed drafts of the article, and approved the final draft.

Yan Wang analyzed the data, authored or reviewed drafts of the article, and approved the final draft.

Linchong Ma analyzed the data, authored or reviewed drafts of the article, and approved the final draft.

Qingxin Li analyzed the data, authored or reviewed drafts of the article, and approved the final draft.

The following information was supplied regarding data availability:

This article is a literature review.

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
