# Peer review of "Relationship between ferroptosis and mitophagy in acute lung injury: a mini-review"

_PeerJ, doi:10.7717/peerj.18062_

## Round 0.1 · original submission · Major Revisions

Dear authors,

I have now received the reviewer reports from 3 experts in the field. Post review of your manuscript myself, I share the opinion of Reviewer #2 that your Review Article requires 'Major Revision' before it can be considered further in the review system.

Can I please therefore ask that you prepare a substantially improved version of your originally submitted article which addresses the concerns and comments raised by Reviewer #2, as well those concerns / comments raised by Reviewers #1 and #3.

Upon re-submission of your revised article, please also include a "Letter of Response to reviewers' Comments" document that clearly outlines, via a point-by-point approach, how you addressed each reviewer concern / comment in the revised article submission.

I look forward to receiving your revised article in the near future.

All the best,
Andrew Eamens

Reviewer 1 ·

Basic reporting

This review article summarizes the relationship between mitophagy and ferroptosis in acute lung injury (ALI). The literature already showed a couple of reviews and research papers reporting the relation between mitophagy and ferroptosis in various organs such as the heart, brain, and testis.
Although it is well written, the figure and table need to be improved. An additional figure showing how mitophagy impacts ferroptosis is required. The table is not clear, and the font needs to be adjusted.
An additional paragraph showing the effect of ethanol on mitophagy and ferroptosis could be added to this review.

Experimental design

The review design is fine as a narrative review but needs improvement as shown above.

Validity of the findings

I think this review requires major improvements related to the figures and tables.

·

Basic reporting

a. The language in the article is generally clear and simple. In general, the article text is technically correct.
b. The review is within the scope of the journal and is broad and cross-disciplinary
c. This field has been reviewed recently, especially in the following:
Liu X, Zhang J, Xie W. The role of ferroptosis in acute lung injury. Mol Cell Biochem. 2022 May;477(5):1453-1461. doi: 10.1007/s11010-021-04327-7. Epub 2022 Feb 15. PMID: 35166985; PMCID: PMC8853161.
However, the authors add more focus on mitophagy than in other reviews. The authors, though, should look at reviews similar to the one above and discuss how such efforts are different than the goals of their review.
d. The introduction is clear and does a good job to motivate the topic. However, the three paragraphs of the introduction would be improved by linking them. As is, the paragraph transitions seem abrupt.

Experimental design

a. The survey methodology seems comprehensive
b. Sources are adequately cited. It would be great if authors can include more references in the beginning and end of the introduction.
c. The review is well organized into appropriate sections and subsections

Validity of the findings

a. The authors should build a stronger argument in the introduction to clearly set out their goals and perspective.
b. The conclusion should include more discussion of future directions and significant remaining unresolved questions.

Additional comments

- Some of the references are not imported correctly (e.g., line 257)
- The figure is not referred to correctly in the manuscript
- Another figure is needed to summarize ferroptosis and its relation to ALI. It would be also great if the authors can include a figure to summarize ALI in general.
- Figure caption should be added to thoroughly describe each section of the figure.

·

Basic reporting

It seems OK.

Experimental design

no comment

Validity of the findings

no comment

Additional comments

-It is noted that Figure 1 and Table 1 are not referenced in the text. Furthermore, Table 1 should be provided for both Ferroptosis and Autophagy.
-There is no figure 3 mentioned in 107.
-Figure 1 is confusing and needs to be sectioned properly and provided with more detailed description and abbreviation list. In addition, it is better to show the pathways or molecules connecting Ferroptosis and Autophagy in general and in ALI.
the reference in line 678 should be rewritten in English.

---

## Round 0.2 · Minor Revisions

Dear authors,

Thank you for addressing the concerns raised in the original round of review in your revised manuscript.

I have now reviewed the revision myself, and I am asking for Minor Revisions.
Please find the annotated copy of your review article attached for your use in the new round of revision.

Kind regards,
Andrew Eamens

·

Basic reporting

no comment

Experimental design

no comment

Validity of the findings

no comment

Additional comments

no comment

---

## Round 0.3 · accepted · Accept

Dear authors,
Thank you kindly for addressing all remaining concerns with the revised version of your original submission.
I think this subsequent round of review as greatly increased the standard of the review.
I did not re-invite the original peer reviewers for this round of review as all requested changes were made by me, and you have addressed all of my concerns/comments.
In my opinion your Review article is at a standard ready for publication acceptance in PeerJ.
Thank you for working through the review round so efficiently, and congratulations on having your article accepted for publication.
All the best,
Andrew Eamens